# Effects of blood meal source and seasonality on reproductive traits of *Culex quinquefasciatus* (Diptera: Culicidae)

Kevin Alen Rucci[1,2]*, Gabriel Barco[1,2], Andrea Onorato[1,2], Mauricio Beranek[1], Mariana Pueta[2,3], Adrián Díaz[1,2]*

[1]Laboratorio de Arbovirus, Instituto de Virología "Dr. J. M. Vanella" (InViV), Facultad de Ciencias Médicas (FCM), Universidad Nacional de Córdoba (UNC), Córdoba, Argentina; [2]Consejo Nacional de Investigaciones Científicas y Técnicas (CONICET), Ciudad Autónoma de Buenos Aires, Argentina; [3]Instituto de Investigaciones en Biodiversidad y Medioambiente (INIBIOMA), CONICET - Universidad Nacional de Comahue (UNCo), San Carlos de Bariloche, Argentina

*For correspondence:
kevin.rucci.224@gmail.com (KAR);
adrian.diaz@fcm.unc.edu.ar (AD)

**Competing interest:** The authors declare that no competing interests exist.

## eLife Assessment

This **useful** study provides the first assessment of potentially interactive effects of seasonality and blood source on mosquito fitness, together in one study. During revision, the manuscript has been substantively improved, providing additional **solid** data to support the robustness of observations. Overall, this interesting study will advance our current understanding of mosquito biology.

**Abstract** Host selection by mosquitoes is a keystone in understanding viral circulation and predicting future infection outbreaks. *Culex* mosquitoes frequently feed on birds during spring and early summer, shifting into mammals towards late summer and autumn. This host switch may be due to changes in mosquito fitness. The aim of this study was to assess if the interaction effect of blood meal source and seasonality may influence reproductive traits of *Culex quinquefasciatus* mosquitoes. For this purpose, *Cx. quinquefasciatus* mosquitoes were reared in simulated summer and autumn conditions and fed on two different hosts, chickens and mice, in a factorial design. Fecundity, fertility, and hatchability during two consecutive gonotrophic cycles were estimated. We found greater fecundity and fertility for mosquitoes fed upon birds than mammals. Fecundity and fertility did not vary between seasons for chicken-fed mosquitoes, whereas in autumn they decreased for mouse-fed mosquitoes. These traits decreased in the second gonotrophic cycle for mouse-fed mosquitoes, whereas they did not vary between cycles for chicken-fed mosquitoes. There was no statistically significant effect of blood meal source, seasonality or their interaction on hatchability, hence this variable was similar among treatments. Overall, these results indicate a statistically significant interaction effect of blood meal source and seasonality on fecundity and fertility. However, the pattern was opposite in relation to our hypothesis, suggesting that further studies are needed to confirm and expand our knowledge about mosquito biology and its relationship with seasonal host use shifting.

## Introduction

The southern house mosquito, *Culex quinquefasciatus* Say (Diptera: Culicidae), is a worldwide distributed vector of numerous pathogens, including *Wuchereria bancrofti* Cobbold (*Kasili et al., 2009*), *Dirofilaria immitis* Leidy (*Labarthe et al., 1998*), and avian malaria (*Farajollahi et al., 2011*). Besides, this mosquito is also one of the main species responsible for the transmission of the arboviruses West Nile (WNV) and St. Louis encephalitis (SLEV) viruses (*Farajollahi et al., 2011*). Because of this, this species is of critical concern due to its impact on both public and veterinary health.

Since arboviruses are transmitted by the bite of infected mosquitoes between vertebrate hosts, activity of these pathogens is determined, at least in part, by the blood feeding habits of these vectors. According to its host feeding pattern range, *Culex* mosquitoes can be regarded as specialists or generalists. Specialist species include mosquitoes feeding primarily on mammals, birds, or reptiles and amphibians. Generalists are mosquitoes fed on the most abundant available hosts (*Fikrig and Harrington, 2021*). Additionally, some species could experience a seasonal shift in their feeding habits, from avian to mammal hosts (*Edman and Taylor, 1968*; *Hancock and Camp, 2022*), being of greater concern from an epidemiological perspective given that they can act as bridge vectors in the transmission of some arboviruses, such as SLEV and WNV (*Kilpatrick et al., 2006*).

West Nile virus (WNV) and St. Louis encephalitis virus (SLEV) are zoonotic pathogens maintained in nature through an enzootic network involving transmission among viremic birds and ornithophagic mosquitoes (*Kilpatrick et al., 2006*). However, during late summer and early autumn, these viruses increase their activity in humans and other mammals, which has led to repeated epidemics in the US (*Kilpatrick et al., 2006*) and South America (*Spinsanti et al., 2008*). Likewise, several *Culex* species, including *Cx. quinquefasciatus*, have exhibited a seasonal shift in their feeding behaviour, feeding primarily on birds in spring and early summer and increasing the proportion of mammal blood meals in late summer and autumn (*Thiemann et al., 2011*). This seasonal shift suggests that viral activity might spill over from birds to mammals as a result of the mosquito host switch (*Edman and Taylor, 1968*; *Kilpatrick et al., 2006*).

Two main contrasting hypotheses have been proposed to explain why primarily ornithophagic mosquitoes increase their feeding on mammals later in the season. The *migration* hypothesis, proposed by *Kilpatrick et al., 2006* for *Cx. pipiens*, suggests that this shift in feeding pattern is related to the abundance of the preferred avian host, the American robin (*Turdus migratorius* L.). This hypothesis is based on their findings that as robin densities decrease, the proportion of mammalian blood meals in *Cx. pipiens* populations increase. However, this hypothesis has several limitations: (i) multiple studies have shown that American robins are not always the preferred host for all *Cx. pipiens* populations (*Apperson et al., 2002*; *Patrican et al., 2007*); (ii) in some areas (e.g. Tennessee) with documented seasonal host shifts, American robins are present year-round (*Savage et al., 2007*) and (iii) this explanation applies only to mosquitoes whose primary host is the American robin, a species found only in the Northern Hemisphere, or another migratory bird. Therefore, it might not be applicable to southern regions, such as Argentina, where WNV seasonality is documented (*Spinsanti et al., 2008*) but American robin is absent. Because of this, it seems that the hypothesis proposed by *Kilpatrick et al., 2006* might be habitat dependent (*Caranci, 2010*). On the other hand, the *defensive behaviour* hypothesis proposed by *Burkett-Cadena et al., 2011*, suggests that host breeding cycles drive the host shift of mosquitoes. According to this hypothesis, in hosts with parental care, during reproductive seasons there is a greater investment of energy in assuring the offspring survivorship, consequently producing an increase in susceptibility of being bitten by mosquitoes as a result of a decrease in defensive behaviours. This event leads to the detection of peaks of host use during periods of reproductive investment, in summer for birds and autumn for mammals (*Burkett-Cadena et al., 2011*). While this hypothesis was proposed as an alternative to address the limitations of the migration hypothesis, it also has drawbacks. It is based on a limited range of host species, assumes a distinct seasonal phenology—which is not present for all mosquito hosts—and remains untested, leaving its validity uncertain.

Explanations for the seasonal shift in mosquito host feeding focused on vector biology are largely lacking in the literature. Numerous biological factors, such as stress, metabolic rate, and blood meal source, along with environmental variables like temperature and photoperiod, influence mosquito physiology and can affect reproductive traits such as fecundity, development rate, and survivorship. These factors can give rise to new nutritional requirements that may ultimately lead to seasonal

variations in host selection by mosquitoes (*Ciota et al., 2014*; *Costanzo et al., 2015*; *Gervasi et al., 2016*; *Yan et al., 2018*; *Yan et al., 2017*). While several studies have examined the effects of these biological and environmental variables on mosquito reproduction, the complex nature of mosquito biology suggests that multiple interactions among these variables may produce novel responses that are not apparent when considered individually. The aim of this study was to assess whether there is an interaction effect between the source of the host blood meal and seasonality (defined by temperature and photoperiod) on three reproductive traits of *Culex quinquefasciatus* mosquitoes: fecundity, fertility, and hatchability. We hypothesize that the interaction between these two variables influences the reproductive outcomes, potentially leading to a seasonal shift in host selection, driven by a reproductive advantage. Given the reported seasonal changes in host use by *Cx. quinquefasciatus*, in autumn we expect a greater number of eggs (fecundity) and larvae (fertility) in mosquitoes after feeding on a mammal host compared to an avian host, and the opposite trend in summer.

## Results

A total of 1162 egg rafts were obtained for analysis, distributed across the eight treatments as follows: chicken:autumn-I: 260, chicken:autumn-II: 174, chicken:summer-I: 162, chicken:summer-II: 88, mouse:autumn-I: 146, mouse:autumn-II: 96, mouse:summer-I: 167, and mouse:summer-II: 69. The full

**Table 1.** Reproductive outcomes of *Culex quinquefasciatus* indicating average fecundity, fertility and hatchability for both gonotrophic cycles, accompanied with its respective standard deviation.

| Replicate | Blood source | Season | Gonotrophic cycle | No. engorged females | No. egg rafts | Fecundity (eggs/ raft) | Fertility (larvae/ raft) | Hatchability (larvae/eggs) |
|---|---|---|---|---|---|---|---|---|
| 1 | chicken | autumn | I | 163 | 76 | 147.47±31.84 | 133.24±38.19 | 0.89±0.14 |
| 1 | chicken | summer | I | 116 | 74 | 125.64±39.57 | 109.39±43.66 | 0.85±0.19 |
| 1 | mouse | autumn | I | 58 | 23 | 124.48±20.85 | 119.96±21.47 | 0.96±0.04 |
| 1 | mouse | summer | I | 208 | 120 | 150.67±29.41 | 140.38±33.15 | 0.93±0.10 |
| 1 | chicken | autumn | II | 73 | 36 | 142.14±36.59 | 129.09±43.62 | 0.87±0.18 |
| 1 | chicken | summer | II | 38 | 27 | 138.23±30.27 | 117.27±37.99 | 0.85±0.18 |
| 1 | mouse | autumn | II | 23 | 15 | 89.13±19.33 | 73.29±28.09 | 0.84±0.24 |
| 1 | mouse | summer | II | 92 | 45 | 106.68±31.97 | 99.16±31.18 | 0.93±0.10 |
| 2 | chicken | autumn | I | 178 | 89 | 138.23±37.45 | 122.7±41.55 | 0.90±0.15 |
| 2 | chicken | summer | I | 78 | 51 | 128.75±38.9 | 126.08±33.97 | 0.89±0.13 |
| 2 | mouse | autumn | I | 110 | 69 | 103.04±19.01 | 91.32±22.96 | 0.89±0.16 |
| 2 | mouse | summer | I | 36 | 27 | 113.26±30.66 | 103.83±34.81 | 0.89±0.20 |
| 2 | chicken | autumn | II | 86 | 66 | 127.4±37.07 | 116.35±39.9 | 0.89±0.16 |
| 2 | chicken | summer | II | 48 | 34 | 143.68±28.06 | 136.75±28.05 | 0.92±0.11 |
| 2 | mouse | autumn | II | 63 | 52 | 88.87±19.65 | 77.86±22.88 | 0.88±0.15 |
| 2 | mouse | summer | II | 18 | 14 | 78.29±25.77 | 67±25.79 | 0.87±0.18 |
| 3 | chicken | autumn | I | 150 | 96 | 147.22±33.4 | 129.33±39.45 | 0.88±0.16 |
| 3 | chicken | summer | I | 67 | 38 | 145.03±44.39 | 136.03±45.49 | 0.90±0.14 |
| 3 | mouse | autumn | I | 92 | 54 | 96.04±23.78 | 85.76±27.45 | 0.89±0.18 |
| 3 | mouse | summer | I | 27 | 20 | 104.3±21.42 | 98.35±23.86 | 0.93±0.09 |
| 3 | chicken | autumn | II | 94 | 73 | 144.07±37.49 | 125.23±39.42 | 0.87±0.17 |
| 3 | chicken | summer | II | 35 | 28 | 156.36±26.71 | 141±39.53 | 0.90±0.18 |
| 3 | mouse | autumn | II | 42 | 29 | 79.66±19.94 | 68.76±23.73 | 0.87±0.21 |
| 3 | mouse | summer | II | 17 | 11 | 100.36±22.82 | 96.1±24.06 | 0.94±0.06 |

**Table 2.** Analysis of deviance table for the generalized linear mixed model examining the effects of blood source, seasonality, gonotrophic cycle, and their interactions on the fecundity (eggs/raft) of *Culex quinquefasciatus* mosquitoes across three replicates.
A random intercept for treatment was included to account for variability among the 24 treatments (variance = 0.006169). Abbreviations: LRT $X^2$ = likelihood-ratio test; df = degrees of freedom.

| Factor | LRT $X^2$ | df | p-value |
|---|---|---|---|
| blood source | 19.22 | 1 | $1.16 \times 10^{-5}$ |
| seasonality | 1.50 | 1 | 0.22 |
| gonotrophic cycle | 0.47 | 1 | 0.49 |
| blood source × seasonality | 5.69 | 1 | 0.02 |
| blood source × gonotrophic cycle | 2.74 | 1 | 0.1 |
| seasonality × gonotrophic cycle | 2.07 | 1 | 0.15 |
| blood source × seasonality × gonotrophic cycle | 1.65 | 1 | 0.20 |

count of egg rafts for each replicate (24 treatments), along with the means for fecundity, fertility, and hatchability, are summarized in *Table 1*. This sample size was sufficient to detect the observed effect with a statistical power of 0.8, indicating an 80% chance of detecting a true effect if it exists.

In both models—fecundity and fertility—the interaction between blood source and seasonality was statistically significant (fecundity: LRT $X^2$=5.69, p<0.05; fertility: LRT $X^2$=4.37, p<0.05; *Tables 2 and 3*; *Figures 1A and 2*, *Figure 1—figure supplement 1*, *Figure 2—figure supplement 1*). Mosquitoes that fed on chicken blood had the highest fecundity (summer: 136.4 eggs/raft, autumn: 141.5 eggs/raft) and fertility (summer: 123.9 larvae/raft, autumn: 126 larvae/raft) in both seasons. In summer, fecundity and fertility were 7% and 11% higher, respectively, in chicken-fed mosquitoes compared to mouse-fed mosquitoes. In autumn, these differences increased to 46% for fecundity and 52% for fertility in favor of chicken-fed mosquitoes. No significant seasonal variation was observed in fecundity and fertility for chicken-fed mosquitoes, but both measures were statistically significant lower by 24% and 25%, respectively, in autumn compared to summer for mouse-fed mosquitoes.

In the fertility model, there was also a significant interaction between blood source and gonotrophic cycle (LRT $X^2$=3.98, p<0.05), indicating that the impact of blood type on larvae production differed between the first and second cycles (*Table 3*, *Figure 2B*). Chicken-fed mosquitoes showed no difference in fertility between cycles, while those fed on mouse blood had 25% lower fertility in the second cycle compared to the first. During the second cycle, fertility was 50% higher in chicken-fed mosquitoes compared to mouse-fed mosquitoes.

**Table 3.** Analysis of deviance table for the generalized linear mixed model examining the effects of blood source, seasonality, gonotrophic cycle, and their interactions on the fertility (larvae/raft) of *Culex quinquefasciatus* mosquitoes across three replicates.
A random intercept for treatment was included to account for variability among the 24 treatments (variance = 0.007226). Abbreviations: LRT $X^2$ = likelihood-ratio test; df = degrees of freedom.

| Factor | LRT $X^2$ | df | p-value |
|---|---|---|---|
| blood source | 13.30 | 1 | 0.0026 |
| seasonality | 0.36 | 1 | 0.54 |
| gonotrophic cycle | 0.28 | 1 | 0.59 |
| blood source × seasonality | 4.37 | 1 | 0.03 |
| blood source × gonotrophic cycle | 3.98 | 1 | 0.04 |
| seasonality × gonotrophic cycle | 0.92 | 1 | 0.34 |
| blood source × seasonality × gonotrophic cycle | 0.50 | 1 | 0.47 |

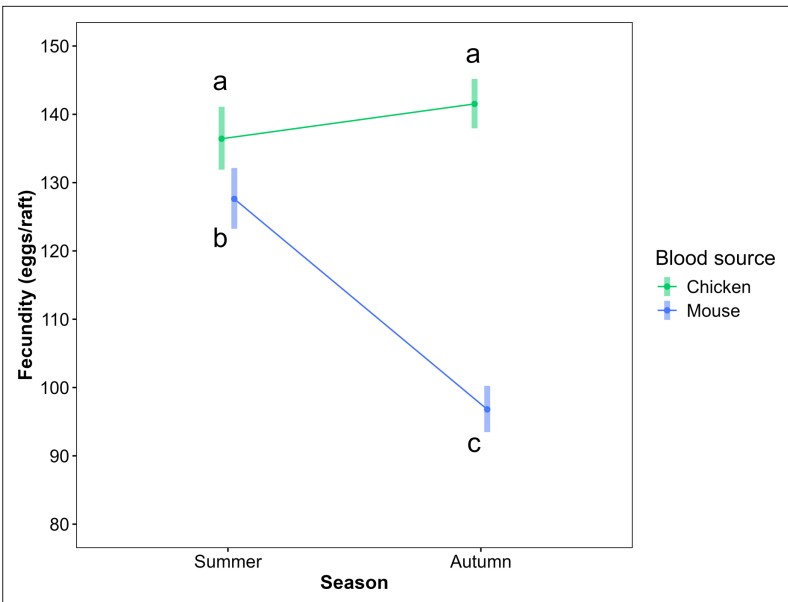

**Figure 1.** Interaction plot for fecundity model. Interaction plot showing the effect of blood meal source (chicken or mouse) and seasonality (autumn or summer) on fecundity (eggs/raft) of *Culex quinquefasciatus*. Points represent the predicted marginal mean ± 95% confidence intervals, obtained from a generalized linear mixed model (GLMM) with a negative binomial distribution and log link function, including treatment as a random intercept effect. Means sharing same letters are not statistically different (p>0.05), based on pairwise comparisons with Tukey's adjustment. Treatments: Summer-Chicken (n = 250), Summer-Mouse (n = 236), Autumn-Chicken (n = 434), Autumn-Mouse (n = 242).

The online version of this article includes the following figure supplement(s) for figure 1:

**Figure supplement 1.** Variation of statistical power of the interaction between blood source and seasonality for the fecundity model across several sample sizes.

*Figure 3* presents the predicted mean values of fecundity and fertility for the 24 treatments, with no differences observed across the three replicates. The variance of random intercepts among treatments was approximately 0.006 for fecundity and 0.007 for fertility, accounting for 6% and 4% of the total variance in the respective models.

There was no statistical difference among treatments and replicates in terms of hatchability, given that no statistical effects of blood source, seasonality or gonotrophic cycle were observed (*Table 4*). The mean hatchability for all treatments was 0.89, with a range between 0.84 for the mouse:autumn-II (replicate 1) colony and 0.96 for mouse:autumn-I (replicate 1; *Table 1*, *Figure 4*).

## Discussion

Understanding the factors influencing mosquito host selection is crucial for predicting arbovirus transmission dynamics. Seasonal shift in host utilization have been widely attributed to changes in host behaviour (*Burkett-Cadena et al., 2011*; *Kilpatrick et al., 2006*), yet the role of mosquito biology in this process remains largely unexplored. Blood meal source and seasonality are known to affect mosquito reproductive success, but their combined effect has received little attention. Our study tested the hypothesis that the interaction between blood meal source and seasonality could affect mosquito fitness, a key factor that could play a role in seasonal host shift. By examining this interaction, we aimed to assess whether physiological processes like reproduction could be considered a candidate for shaping seasonal feeding patterns in *Culex quinquefasciatus*.

If seasonal host shift were driven by fitness advantages, we would expect mosquitoes to exhibit higher reproductive output when feeding on the predominant host in each season—that is, greater fecundity and fertility with bird blood in summer and with mammal blood in autumn. However, our findings did not fully align with these expectations. While mosquitoes fed on birds maintained high reproductive success in both summer and autumn, those fed on mammals exhibited a significant

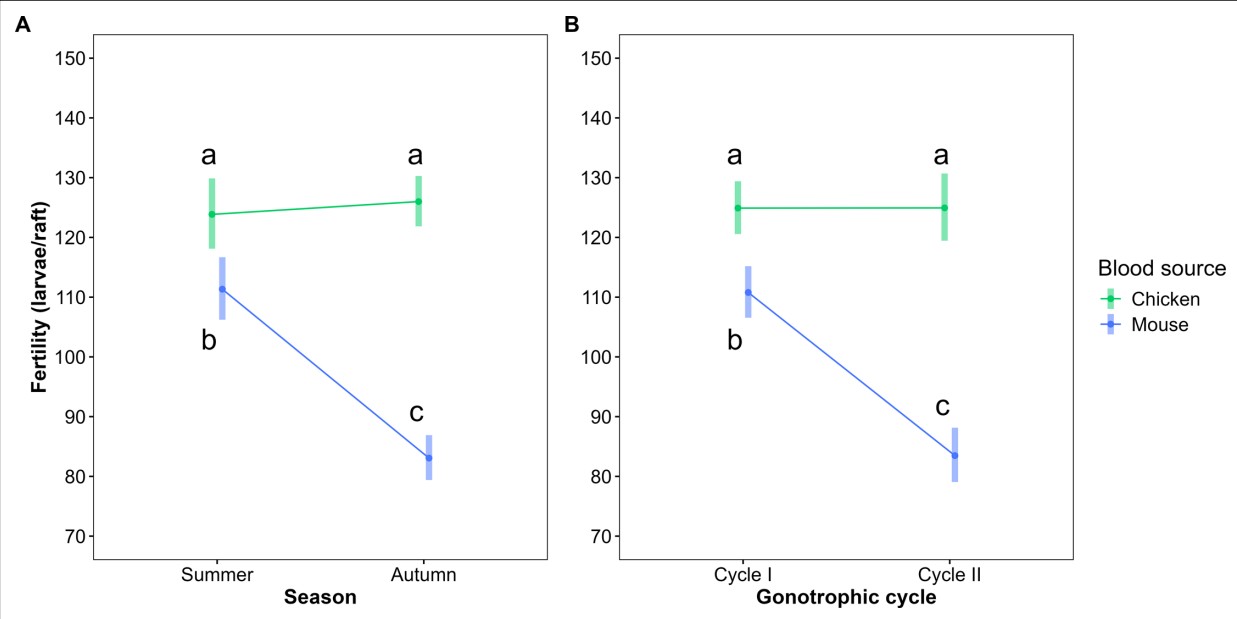

**Figure 2.** Interaction plots for fertility model. Interaction plots showing the effect of blood meal source (chicken or mouse) and seasonality (autumn or summer) (**A**) or blood meal source and gonotrophic cycle (first or second) (**B**) on fertility (eggs/raft) of *Culex quinquefasciatus*. Points represent the predicted marginal mean ± 95% confidence intervals, obtained from a generalized linear mixed model (GLMM) with a negative binomial distribution and log link function, including treatment as a random intercept effect. Means sharing same letters are not statistically different (p>0.05), based on pairwise comparisons with Tukey's adjustment. Treatments: Summer-Chicken: (n = 250), Summer-Mouse (n = 236), Autumn-Chicken (n = 434), Autumn-Mouse (n = 242); Cycle I-Chicken (n = 422), Cycle I-Mouse (n = 313), Cycle II-Chicken (n = 262), Cycle II-Mouse (n = 165).

The online version of this article includes the following figure supplement(s) for figure 2:

**Figure supplement 1.** Variation of statistical power of the interaction between blood source and seasonality for the fertility model across several sample sizes.

decline in reproductive output during autumn. This suggests that, rather than conferring a reproductive advantage in this season, mammalian blood may become a suboptimal resource under autumn conditions, potentially due to metabolic constraints or differences in nutrient assimilation.

The effect of blood meal on fitness observed in this study (greater in bird-fed mosquitoes) is consistent with previous findings on *Cx. quinquefasciatus* (**Richards et al., 2012**; **Telang and Skinner, 2019**) and other *Culex* (**Demirci et al., 2014**) and *Aedes* (**Harrison et al., 2021**) species. Avian blood is known to be nutritionally richer than mammalian blood, particularly in proteins and essential amino acids (e.g. isoleucine) required for egg development (**Alto et al., 2014**). Beyond intrinsic differences in blood composition, environmental factors such as temperature and photoperiod (seasonality) may further modulate how nutrients from a blood meal are allocated. While most of the nutrients obtained from a blood meal are directed toward egg production, a portion is also required for non-reproductive physiological functions (**Rivera-Pérez et al., 2017**), and the extent to which these resources are utilized may depend on external conditions such as temperature (**Briegel and Timmermann, 2001**).

Previous studies have shown that temperature regulates oogenesis and influences how mosquitoes allocate proteins obtained from a blood meal. In *Aedes aegypti*, for example, the proportion of blood-derived proteins allocated to egg production varies with temperature: at low temperatures, approximately 50% of these proteins are used for oogenesis, whereas at higher temperatures, this proportion drops to 25% (**Briegel and Timmermann, 2001**). If a similar mechanism occurs in *Cx. quinquefasciatus*, it could partially explain why mammal-fed mosquitoes exhibited a more substantial reproductive decline in autumn. Under lower temperatures, a greater proportion of proteins may be required for egg development, which could disproportionately impact individuals feeding on nutritionally poorer blood sources.

In addition to this blood meal-seasonal effect, we also found a significant interaction between blood meal source and gonotrophic cycle. This result suggests that the impact of blood sources is not uniform across cycles. If blood meal source were the only determinant of reproductive success,

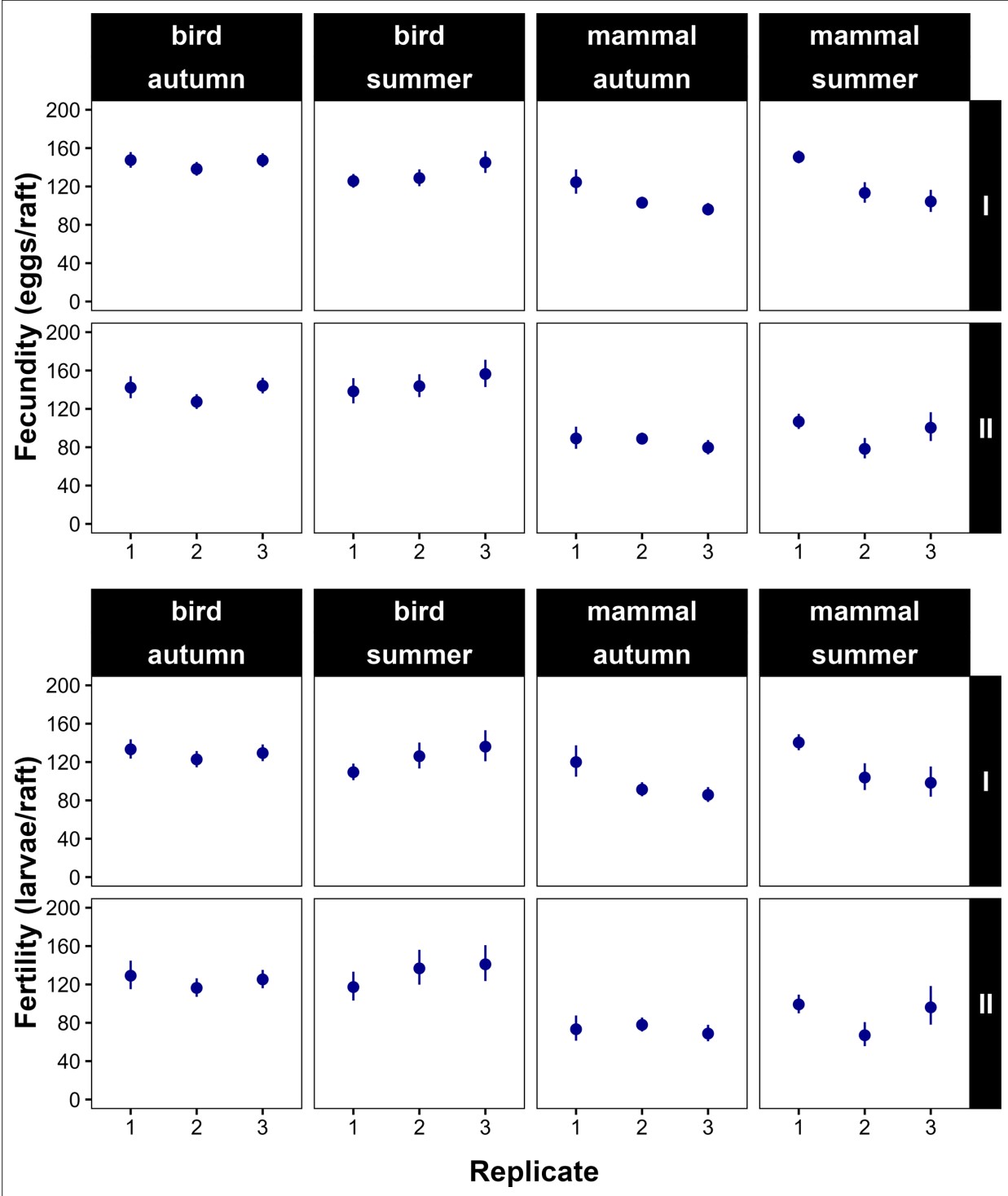

**Figure 3.** Variation of reproductive outputs across replicates. Dotplots showing the variation of predicted fecundity (eggs/raft) and fertility (larvae/raft) means ± 95% confidence intervals of *Culex quinquefasciatus* along the 24 treatments, with combination of blood source (chicken or mouse), seasonality (autumn or summer), gonotrophic cycle (first or second) and replicate (1–3), obtained from a generalized linear mixed model (GLMM) with a negative binomial distribution and log link function, including treatment as a random intercept effect. Treatments: Summer-Chicken-I (n = 162), Summer-Chicken-II (n = 88), Summer-Mouse-I (n = 167), Summer-Mouse-II (n = 69), Autumn-Chicken-I (n = 260), Autumn-Chicken-II (n = 174), Autumn-Mouse-I (n = 146), Autumn-Mouse-II (n = 96).

**Table 4.** Analysis of deviance table for the generalized linear model examining the effects of blood source, seasonality, gonotrophic cycle, replicate and their interactions on the hatchability (larvae/eggs) of *Culex quinquefasciatus* mosquitoes.
Abbreviations: LRT $X^2$ = likelihood-ratio test; df = degrees of freedom.

| Factor | LRT $X^2$ | df | p-value |
|---|---|---|---|
| blood source | 2.50 | 1 | 0.11 |
| seasonality | 3.06 | 1 | 0.08 |
| gonotrophic cycle | 0.005 | 1 | 0.94 |
| replicate | 1.31 | 1 | 0.51 |
| blood source × seasonality | 0.12 | 1 | 0.72 |
| blood source × gonotrophic cycle | 3.64 | 1 | 0.06 |
| seasonality × gonotrophic cycle | 0.07 | 1 | 0.79 |
| blood source × replicate | 2.42 | 1 | 0.30 |
| seasonality × replicate | 2.96 | 1 | 0.23 |
| gonotrophic cycle × replicate | 0.07 | 1 | 0.96 |
| blood source × seasonality × gonotrophic cycle | 2.73 | 1 | 0.10 |
| blood source × seasonality × replicate | 0.02 | 1 | 0.99 |
| blood source × gonotrophic cycle × replicate | 2.48 | 1 | 0.29 |
| seasonality × gonotrophic cycle × replicate | 0.43 | 1 | 0.81 |
| blood source × seasonality × gonotrophic cycle × replicate | 2.57 | 1 | 0.28 |

we would expect its effects to be consistent across cycles, with mosquitoes fed on birds maintaining higher reproductive output than those fed on mammals in a stable pattern. However, our findings showed that while bird-fed mosquitoes sustained similar reproductive output across cycles, those fed on mammals exhibited a significant decline in fertility in the second cycle. Since each cycle involves a new blood meal, this suggests that factors beyond blood nutritional quality contribute to the observed pattern, disproportionately affecting mosquitoes feeding on mammals in later cycles.

A decline in fertility across gonotrophic cycles has also been documented in other *Cx. quinquefasciatus* populations fed on both birds and mammals (*Richards et al., 2012*; *Telang and Skinner, 2019*), suggesting that reproductive output varies not only with seasonality but also with the number of successive cycles. One potential explanation for this decline is age-related reproductive constraints. With each successive cycle, ovarian follicular degeneration increases, leading to a reduction in the number of viable eggs per raft (*Awahmukalah and Brooks, 1985*). Additionally, given that several days separate the first and second cycles, physiological aging could contribute to this pattern (*McCann et al., 2009*). Interestingly, bird-fed mosquitoes maintained stable fertility across cycles, a pattern that has also been observed in previous studies (*Bennett, 1970*). While fertility often declines in later cycles, some studies have reported no change or even an increase in egg production under certain conditions (*Bennett, 1970*). This variability suggests that the effects of successive gonotrophic cycles on fertility may depend on multiple factors, including blood meal quality. Since avian blood is known to be nutritionally richer than mammalian blood, mosquitoes feeding on birds may have access to sufficient resources to sustain egg production across cycles, even as they age. However, further research is needed to determine whether specific nutrient components play a role in maintaining reproductive output over time. Similarly, body size has been shown to influence reproductive patterns in some mosquito species. In *Aedes aegypti*, for instance, larger females exhibit a rapid decline in fecundity after the first cycle, whereas smaller females maintain consistent egg production across early cycles before experiencing a significant drop (*Briegel et al., 2002*). While we did not assess body size in this study, these findings underscore the complexity of reproductive trade-offs in mosquitoes and the importance of considering multiple interacting factors when examining reproductive dynamics across cycles.

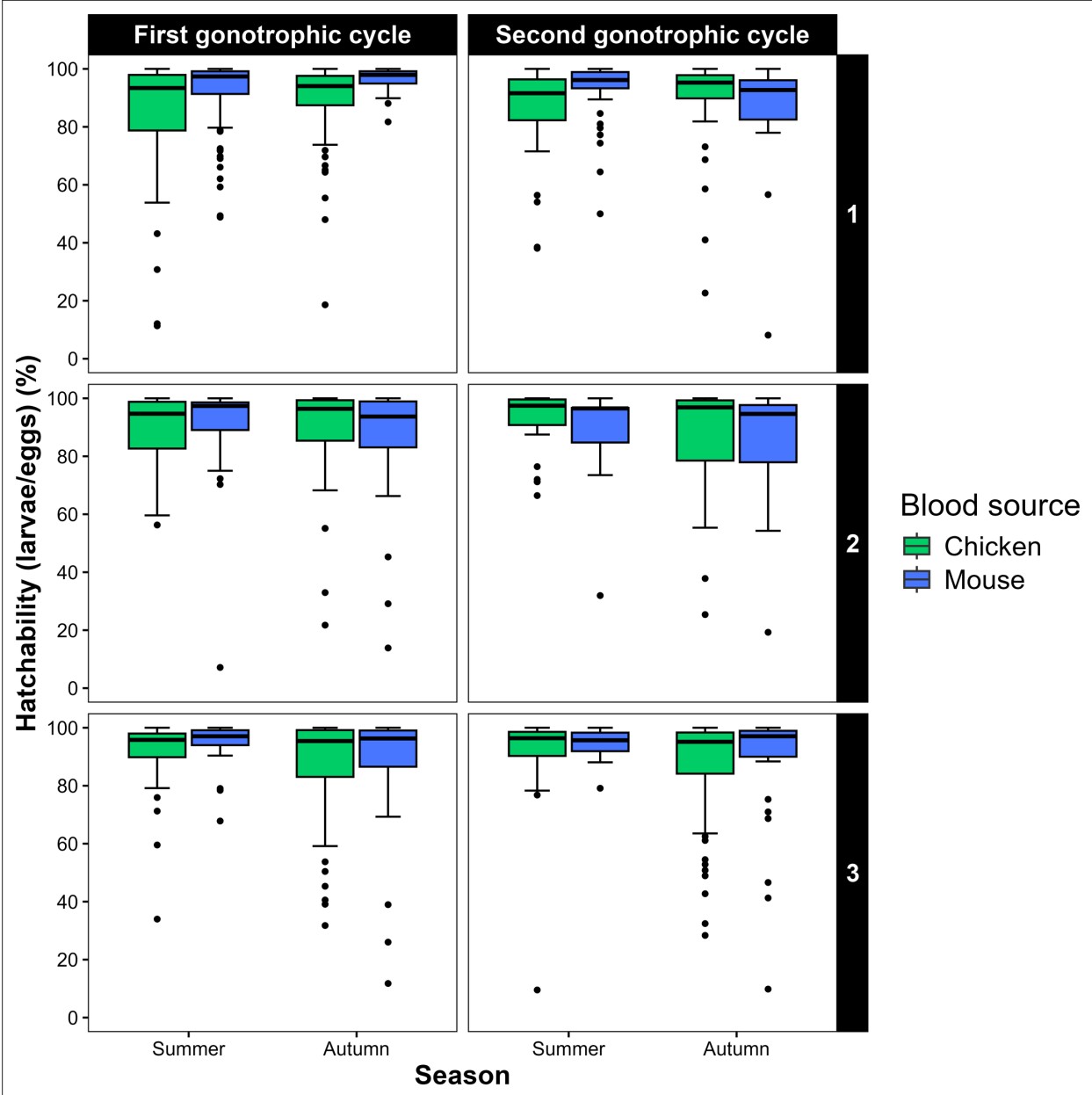

**Figure 4.** Variation of hatchability across replicates. Boxplots showing predicted hatchability median of *Culex quinquefasciatus* for the 24 treatments, combining replicate (1-3), blood meal source (chicken or mouse), seasonality (summer or autumn), and gonotrophic cycle (first or second) . Values obtained from a generalized linear model (GLM) with a quiasipoisson distribution and log link function. Treatments: Summer-Chicken-I (n = 162), Summer-Chicken-II (n = 88), Summer-Mouse-I (n = 167), Summer-Mouse-II (n = 69), Autumn-Chicken-I (n = 260), Autumn-Chicken-II (n = 174), Autumn-Mouse-I (n = 146), Autumn-Mouse-II (n = 96).

Together, these results indicate that both blood meal source and seasonality influence mosquito reproductive output. Moreover, hatchability did not differ among treatments, indicating that differences in egg viability did not drive these effects. This reinforces the idea that the main differences in reproductive output stem from factors affecting fecundity and fertility rather than post-oviposition processes.

While our findings confirm an interaction effect between blood meal source and seasonality, several caveats must be considered. First, our study was conducted under controlled laboratory conditions, which may not fully capture the environmental variability experienced by mosquitoes in the field. Factors such as fluctuating temperature, natural light cycles, and variable humidity levels could influence mosquito reproductive performance in ways not accounted for in our setup. However, as this is

the first study of its kind, future research should aim to incorporate more ecologically realistic conditions. A second limitation commonly encountered in ecological and behavioural studies, is determining an appropriate sample size for detecting biologically meaningful effects (*Taborsky, 2010*). To address this, we repeated the assay across three replicates and conducted a post-hoc power analysis, which suggested that our sample size was adequate given the presence of interaction effects among explanatory variables. Finally, it remains possible that *Cx. quinquefasciatus* populations in Argentina do not exhibit the same host-switching behaviour observed in US populations, potentially influencing the relevance of our findings for different ecological contexts. However, previous studies have reported seasonal shifts in host selection and arbovirus transmission patterns in Argentina (*Beranek, 2019*; *Spinsanti et al., 2008*), suggesting that similar dynamics may be at play. More field-based studies will be essential to assess seasonal host variation in *Cx. quinquefasciatus* and its implications for arbovirus transmission.

In summary, our study investigates a critical aspect of mosquito reproductive success and its potential link to seasonal host selection in *Culex* mosquitoes, a group of significant epidemiological importance due to their role as bridge vectors between avian and human hosts. While we detected a significant interaction between blood meal source and seasonality, our findings do not support the idea that reproductive fitness alone explains seasonal host shifts. Instead, they suggest that the relationship between mosquito reproduction and host selection is more complex, likely influenced by multiple ecological and physiological factors beyond fitness constraints. Given this complexity, alternative explanations for seasonal host shifts should be considered. Genetic predisposition, midgut microbiota composition, physiological constraints, or other biological mechanisms unrelated to fitness may also contribute to shaping host-use patterns (*Gervasi et al., 2016*; *Yan et al., 2018*). Investigating these aspects will be crucial to fully understand the drivers of host selection in mosquitoes. Future research should aim to unravel the intricate relationship between feeding patterns, reproductive physiology, and environmental conditions. Refining our understanding of these dynamics will improve vector ecology models and enhance predictions of how seasonal variations in host selection may influence pathogen transmission.

## Materials and methods
### Establishment and maintenance of mosquitoes
Egg rafts of *Culex quinquefasciatus* were collected from a drainage ditch at Universidad Nacional de Córdoba Campus, Córdoba city, in February 2021. Each raft was individually maintained in plastic containers with one liter of distilled water. The hatched larvae were fed with 100 mg of liver powder three times per week until pupation. Pupae were then transferred to plastic emerging cages (21 cm x 12 cm) covered with a tulle-like fabric, containing distilled water but no food. Adults emerging from each raft were identified morphologically (*Darsie, 1985*) and molecularly (*Smith and Fonseca, 2004*) to ensure they corresponded to *Cx. quinquefasciatus*, since *Cx. pipiens* and its hybrids coexist sympatrically in Córdoba city (*Branda et al., 2021*). All adults were reared in a cardboard cage of 22.5 L (28 cm x 36.5 cm) and provided *ad libitum* with a 10% sugar solution soaked in cotton pads placed on plastic cups. For long-term maintenance of the colony, 24-hour-starved mosquitoes were offered a blood meal from a restrained chicken twice a month. Four days after feeding, a plastic container with distilled water was placed inside the cage to allow engorged females to lay egg rafts. Batches of egg were collected and transferred to plastic containers (30 cm x 25 cm x 7 cm) in a proportion of three rafts per container, filled with 3 L of distilled water. The hatched larvae were also fed with liver powder at the same proportion described above. Pupae were transferred to emerging cages, and adult mosquitoes were placed in the final cardboard cage. The colony has been maintained for over twenty generations in the Insectary of the Instituto de Virología 'Dr. J.M. Vanella' (InViV). Room conditions were controlled at 28 ± 1 °C, with a 12 L:12D photoperiod and 70% relative humidity.

### Experimental design
#### Blood source
For experimental trials, avian hosts (live chicks of the species *Gallus gallus*) and mammalian hosts (live mice of the species *Mus musculus*, strain C57BL/6) were used to evaluate the effect of blood meal source on fecundity, fertility, and hatchability. Chicks were generously donated by the Bartolucci

poultry farm (Córdoba, Argentina), while mice were commercially obtained from the Instituto de Investigación Médica Mercedes y Martín Ferreyra (CONICET - Universidad Nacional de Córdoba). In each trial, 24-hour-starved adult female mosquitoes were provided with a blood meal from restrained chicks or mice. Vertebrate hosts were offered to mosquitoes one hour before the lights were turned off and were kept for 3 hr during two consecutive gonotrophic cycles.

The experimental use of animals (mosquitoes and vertebrates) was approved by the ethical committee at Facultad de Ciencias Médicas, Universidad Nacional de Córdoba (FCM-UNC) in compliance with the legislation regarding the use of animals for experimental and other scientific purposes (accession code: CE-2022–00518476-UNC-SCT#FCM).

## Seasonality

To assess the effect of seasonality (photoperiod + temperature) on fecundity, fertility, and hatchability, a 'typical' summer and autumn day from Córdoba city was simulated in an incubator, where mosquitoes were housed throughout the entire experiment. Summer conditions were as follows: $T°_{min}$ = 22°C, $T°_{max}$ = 28°C, photoperiod = 14 L:10D. Autumn conditions were characterized by: $T°_{min}$ = 16°C, $T°_{max}$ = 22°C, and photoperiod = 10 L:14D. The humidity levels were maintained at 60–70% for both conditions. Data for simulated conditions were obtained from Climate Data website (https://es.climate-data.org/).

## Feeding trial design

The interaction effect of blood source (chicken or mouse) and seasonality (autumn or summer) was evaluated during two consecutive gonotrophic cycles (I and II). This combination produced eight colonies that were established using egg rafts collected from the maintenance colony: mouse:autumn-I, mouse:autumn-II, chicken:autumn-I, chicken:autumn-II, mouse:summer-I, mouse:summer-II, chicken:summer-I, chicken:summer-II. This set of eight colonies was repeated in triplicate (1-3) at different time points, generating a total of 24 treatments.

The experimental colonies were maintained under controlled conditions to ensure that all adults were of the same size (*Kauffman et al., 2017*). Adult mosquitoes were placed in cardboard cages of 8.3 L (21 cm x 24 cm) and were provided with *ad libitum* access to a 10% sugar solution, which was soaked in cotton pads placed on plastic cups.

Feeding trials were conducted at two time points: 5 days post-emergence (first cycle) and 14 days post-emergence (second cycle). Following each blood meal, female mosquitoes were anesthetized using $CO_2$ and classified as fully engorged (VI Sella's stage), partially fed (I-V Sella's stage), or unfed (I Sella's stage), following the classification by *Santos et al., 2019*. Fully engorged females were counted and separated into a separate cage to complete their gonotrophic cycle. Four days after feeding, a cup containing distilled water was placed inside the cage to facilitate oviposition. After each oviposition cycle, egg rafts were collected, counted, and transferred to a 12-well plate containing 4 mL of distilled water. They were then photographed to subsequently determine the number of eggs per raft. Rafts were maintained in each well until they hatched into L1 larvae, at which point they were preserved with 2 mL of 96% ethanol, and the number of L1 larvae per raft was counted.

*Statistical analysis*. The reproductive outputs measured for each experimental colony were fecundity, fertility, and hatchability. All analyses were performed using R Studio statistic software, v.4.2.1 (*R Development Core Team, 2022*).

*Fecundity* was defined as the number of eggs per raft and *fertility* as the number of L1 larvae hatched per raft. To evaluate the effect of blood meal source and seasonality on these two variables, a Generalized Linear Mixed Model (GLMM; *lme4* package) with negative binomial error distribution and logarithmic link function was adjusted (*Bates et al., 2015*). Fecundity and fertility served as response variables, while the fixed effect variables included blood source, seasonality, and gonotrophic cycle, all in a three-way interaction. Treatment (n = 24) was set as a random effect variable.

*Hatchability* (= hatching rate) was defined as the ratio of the number of larvae to the number of eggs per raft. While the original intent was to apply a GLMM to all three variables, the validation of GLMMs for hatchability across several distributions revealed poor fit. Consequently, a Generalized Linear Model (GLM; MASS package) with a quasipoisson error distribution and logarithmic link function was used instead (*Venables and Ripley, 2002*). In this model, the response variable was the number of larvae, while the explanatory variables included blood source, seasonality, gonotrophic

cycle, and replicate as a four-way interaction. Additionally, the logarithm of the number of eggs was incorporated as an offset in the model.

Multiple comparisons among treatments were conducted using the Tukey's honestly significant difference test (HSD Tukey), incorporating the *Kramer, 1956* correction for unbalanced data, also known as the Tukey-Kramer method. This approach allows to set the family-wise error rate to 0.05. These comparisons were performed using the *emmeans* package (*Lenth, 2024*).

Model validation was assessed using a simulation-based approach to generate randomized quantile residuals to test goodness of fit (Q-Q plot), homoscedasticity, and outliers and influential points (Cook's distance and Leverage plots). These analyses were conducted with the *DHARMa* package (*Hartig, 2022*).

Additionally, post-hoc statistical power was evaluated using the *mixedpower* package, a tool specifically designed for simulating statistical power in Generalized Linear Mixed Models (GLMMs). The analysis proceeded in three key steps: (1) simulation of new datasets based on the parameters of an original fitted model (the fecundity and fertility models), (2) refitting the model to each dataset and testing for statistical significance, and (3) calculating statistical power as the proportion of simulated datasets where the effect of interest reached significance. Fixed effects were simulated for sample sizes ranging from 30 to 150, with 1000 datasets generated at each sample size to ensure robust estimation. The variable treatment was designated as a random effect, capturing variability associated with this factor. For further details on the methodology, including considerations for simulation accuracy and model structure, refer to *Kumle et al., 2021*.

## Acknowledgements

We would like to thank Dr. Agustín Quaglia for providing valuable help in statistical analysis. In addition, we greatly appreciate very helpful comments on the manuscript by anonymous reviewers.

## Additional information

### Funding

| Funder | Grant reference number | Author |
|---|---|---|
| Agencia Nacional de Promoción de la Investigación, el Desarrollo Tecnológico y la Innovación | PICT 2018-1172 | Adrián Díaz |
| Universidad Nacional de Córdoba | Consolidar 2018-2023 | Adrián Díaz |

The funders had no role in study design, data collection and interpretation, or the decision to submit the work for publication.

### Author contributions

Kevin Alen Rucci, Data curation, Software, Formal analysis, Validation, Investigation, Visualization, Methodology, Writing – original draft, Writing – review and editing; Gabriel Barco, Andrea Onorato, Mauricio Beranek, Data curation, Investigation, Writing – review and editing; Mariana Pueta, Conceptualization, Supervision, Validation, Visualization, Methodology, Writing – review and editing; Adrián Díaz, Conceptualization, Supervision, Funding acquisition, Validation, Methodology, Project administration, Writing – review and editing, Resources

### Author ORCIDs

Kevin Alen Rucci ⬩ https://orcid.org/0000-0002-0940-2423
Andrea Onorato ⬩ https://orcid.org/0009-0008-6967-398X
Adrián Díaz ⬩ https://orcid.org/0000-0001-5953-2907

## Ethics

The experimental use of animals (mosquitoes and vertebrates) was approved by the ethical committee at Facultad de Ciencias Universidad Nacional de Córdoba (FCM-UNC) in compliance with the legislation regarding the use of animals for experimental and other scientific purposes (accession code: CE-2022-00518476-UNC-SCT#FCM).

Reviewer #1 (Public review): https://doi.org/10.7554/eLife.89485.5.sa1
Reviewer #2 (Public review): https://doi.org/10.7554/eLife.89485.5.sa2
Author response https://doi.org/10.7554/eLife.89485.5.sa3

## Additional files

### Supplementary files

MDAR checklist

### Data availability

All data generated and analyzed in this study, along with the accompanying code, are publicly available on Figshare (https://doi.org/10.6084/m9.figshare.28555616).

The following dataset was generated:

| Author(s) | Year | Dataset title | Dataset URL | Database and Identifier |
|---|---|---|---|---|
| Rucci KA | 2025 | Database and R code for Rucci et al. (2025) - "Effects of blood meal source and seasonality on reproductive traits of Culex quinquefasciatus (Diptera: Culicidae)" | https://doi.org/10.6084/m9.figshare.28555616 | figshare, 10.6084/m9.figshare.28555616 |

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
