## [Editor Report · eLife Assessment]

This **useful** study provides the first assessment of potentially interactive effects of seasonality and blood source on mosquito fitness, together in one study. During revision, the manuscript has been substantively improved, providing additional **solid** data to support the robustness of observations. Overall, this interesting study will advance our current understanding of mosquito biology.

---

## [Referee Report · Reviewer #1 (Public review)]

This study examines the role of host blood meal source, temperature, and photoperiod on the reproductive traits of Cx. quinquefasciatus, an important vector of numerous pathogens of medical importance. The host use pattern of Cx. quinquefasciatus is interesting in that it feeds on birds during spring and shifts to feeding on mammals towards fall. Various hypotheses have been proposed to explain the seasonal shift in host use in this species but have provided limited evidence. This study examines whether the shifting of host classes from birds to mammals towards autumn offers any reproductive advantages to Cx. quinquefasciatus in terms of enhanced fecundity, fertility, and hatchability of the offspring. The authors found no evidence of this, suggesting that alternate mechanisms may drive the seasonal shift in host use in Cx. quinquefasciatus.

---

## [Referee Report · Reviewer #2 (Public review)]

Conceptually, this study is interesting and is the first attempt to account for the potentially interactive effects of seasonality and blood source on mosquito fitness, which the authors frame as a possible explanation for previously observed host-switching of Culex quinquefasciatus from birds to mammals in the fall. The authors hypothesize that if changes in fitness by blood source change between seasons, higher fitness on birds in the summer and on mammals in the autumn could drive observed host switching. To test this, the authors fed individuals from a colony of Cx. quinquefasciatus on chickens (bird model) and mice (mammal model) and subjected each of these two groups to two different environmental conditions reflecting the high and low temperatures and photoperiod experienced in summer and autumn in Córdoba, Argentina (aka seasonality). They measured fecundity, fertility, and hatchability over two gonotrophic cycles. The authors then used generalized linear mixed models to evaluate the impact of host species, seasonality, and gonotrophic cycle on fecundity, fertility, and hatchability. The authors were trying to test their hypothesis by determining whether there was an interactive effect of season and host species on mosquito fitness. This is an interesting hypothesis; if it had been supported, it would provide support for a new mechanism driving host switching. While the authors did report an interactive impact of seasonality and host species, the directionality of the effect was the opposite from that hypothesized. The authors have done a very good job of addressing many of the reviewer's concerns, especially by adding two additional replicates.

---

## [Author Response]

The following is the authors’ response to the previous reviews

We would like to thank you for your valuable comments and suggestions, which have greatly contributed to improving our manuscript.

We have carefully addressed all the reviewers' suggestions, and detailed responses for each Reviewer are provided at the end of this letter. In summary:

• The Introduction has been revised to provide a more focused discussion on results, toning down the speculative discussion on seasonal host shifts.

• The methodology section has been clarified, particularly the power analysis, which now includes a clearer explanation. The random effects in the models have been better described to ensure transparency.

• The Results section was reorganized to highlight the key findings more effectively.

• The Discussion has been restructured for clarity and conciseness, ensuring the interpretation of the results is clearer and better aligned with the study objectives.

• Minor edits throughout the manuscript were made to improve readability and accuracy.

We hope you find this revised version of the manuscript satisfactory.

**Reviewer #1 (Public review):**
Summary:This study examines the role of host blood meal source, temperature, and photoperiod on the reproductive traits of Cx. quinquefasciatus, an important vector of numerous pathogens of medical importance. The host use pattern of Cx. quinquefasciatus is interesting in that it feeds on birds during spring and shifts to feeding on mammals towards fall. Various hypotheses have been proposed to explain the seasonal shift in host use in this species but have provided limited evidence. This study examines whether the shifting of host classes from birds to mammals towards autumn offers any reproductive advantages to Cx.quinquefasciatus in terms of enhanced fecundity, fertility, and hatchability of the offspring. The authors found no evidence of this, suggesting that alternate mechanisms may drive the seasonal shift in host use in Cx. quinquefasciatus.Strengths:Host blood meal source, temperature, and photoperiod were all examined together.Weaknesses:The study was conducted in laboratory conditions with a local population of Cx. quinquefasciatus from Argentina. I'm not sure if there is any evidence for a seasonal shift in the host use pattern in Cx. quinquefasciatus populations from the southern latitudes.Comments on the revision:Overall, the manuscript is much improved. However, the introduction and parts of the discussion that talk about addressing the question of seasonal shift in host use pattern of Cx. quin are still way too strong and must be toned down. There is no strong evidence to show this host shift in Argentinian mosquito populations. Therefore, it is just misleading. I suggest removing all this and sticking to discussing only the effects of blood meal source and seasonality on the reproductive outcomes of Cx. quin.

Introduction and discussion have been modified, toned down and sticked to discuss the results as suggested.

**Reviewer #1 (Recommendations for the authors):**
Some more minor comments are mentioned below.Line 51: Because 'of' this,

Changed as suggested.

Line 56: specialists 'or' generalists

Changed as suggested.

Line 56: primarily

Changed as suggested.

Line 98: Because 'of' this,

Changed as suggested.

**Reviewer #2 (Public review):**
Summary:Conceptually, this study is interesting and is the first attempt to account for the potentially interactive effects of seasonality and blood source on mosquito fitness, which the authors frame as a possible explanation for previously observed hostswitching of Culex quinquefasciatus from birds to mammals in the fall. The authors hypothesize that if changes in fitness by blood source change between seasons, higher fitness on birds in the summer and on mammals in the autumn could drive observed host switching. To test this, the authors fed individuals from a colony of Cx. quinquefasciatus on chickens (bird model) and mice (mammal model) and subjected each of these two groups to two different environmental conditions reflecting the high and low temperatures and photoperiod experienced in summer and autumn in Córdoba, Argentina (aka seasonality). They measured fecundity, fertility, and hatchability over two gonotrophic cycles. The authors then used generalized linear mixed models to evaluate the impact of host species, seasonality, and gonotrophic cycle on fecundity, fertility, and hatchability. The authors were trying to test their hypothesis by determining whether there was an interactive effect of season and host species on mosquito fitness. This is an interesting hypothesis; if it had been supported, it would provide support for a new mechanism driving host switching. While the authors did report an interactive impact of seasonality and host species, the directionality of the effect was the opposite from that hypothesized. The authors have done a very good job of addressing many of the reviewer's concerns, especially by adding two additional replicates. Several minor concerns remain, especially regarding unclear statements in the discussion.Strengths:(1) Using a combination of laboratory feedings and incubators to simulate seasonal environmental conditions is a good, controlled way to assess the potentially interactive impact of host species and seasonality on the fitness of Culex quinquefasciatus in the lab.(2) The driving hypothesis is an interesting and creative way to think about a potential driver of host switching observed in the field.Weaknesses:(1) The methods would be improved by some additional details. For example, clarifying the number of generations for which mosquitoes were maintained in colony (which was changed from 20 to several) and whether replicates were conducted at different time points.

Changed as suggested.

(2) The statistical analysis requires some additional explanation. For example, you suggest that the power analysis was conducted a priori, but this was not mentioned in your first two drafts, so I wonder if it was actually conducted after the first replicate. It would be helpful to include further detail, such as how the parameters were estimated. Also, it would be helpful to clarify why replicate was included as a random effect for fecundity and fertility but as a fixed effect for hatchability. This might explain why there were no significant differences for hatchability given that you were estimating for more parameters.

The power analysis was conducted a posteriori, as you correctly inferred. While I did not indicate that it was performed a priori, you are right in noting that this was not explicitly mentioned. As you suggested, the methodology for the power analysis has been revised to clarify any potential doubts.

Regarding the model for hatchability, a model without a random effect variable was used, as all attempts to fit models with random effects resulted in poor validation. These points have now been clarified and explained in the corresponding section.

(3) A number of statements in the discussion are not clear. For example, what do you mean by a mixed perspective in the first paragraph? Also, why is the expectation mentioned in the second paragraph different from the hypothesis you described in your introduction?

Changed as suggested.

(4) According to eLife policy, data must be made freely available (not just upon request).

Data and code will be publicly available. The corresponding section was modified.

**Reviewer #2 (Recommendations for the authors):**
Your manuscript is much improved by the inclusion of two additional replicates! The results are much more robust when we can see that the trends that you report are replicable across 3 iterations of the experiment. Congratulations on a greatly improved study and paper! I have several minor concerns and suggestions, listed below:38-39: I think it is clearer to say "no statistically significant effect of season on hatchability of eggs" ... or specify if you are referring to blood or the interaction of blood and season. It isn't clear which treatment you are referring to here.

Changed as suggested.

54-57: This could be stated more succinctly. Instead of citing papers that deal with specific examples of patterns, I would suggest citing a review paper that defines these terms.

Changed as suggested.

83-84: What if another migratory bird is the preferred host in Argentina? I would state this more cautiously (e.g. "may not be applicable...").

Changed as suggested.

95-96: I don't understand what you mean by this. These hypotheses are specifically meant to understand mosquitoes that DO have a distinct seasonal phenology, so I'm not sure why this caveat is relevant. And naturally this hypothesis is host dependent, since it is based on specific host reproductive investments. I think that the strongest caveat to this hypothesis is simply that it hasn't been proven.

Changed as suggested.

97-115: This is a great paragraph! Very clear and compelling.

Thanks for your words!

118: Do you have an exact or estimated number of rafts collected?

Sorry, I have not the exact number of rafts, but it was at leas more than 20-30.

135: "over twenty" was changed to "several"; several would imply about 3 generations, so this is misleading. If the colony was actually maintained for over twenty generations, then you should keep that wording.

Changed as suggested.

163-164: Can you please clarify whether the replicates were conducted a separate time points?

Changed as suggested.

Note: the track changes did not capture all of the changes made; e.g. 163-164 should show as new text but does not.

You are absolutely right; when I uploaded the last version, I unfortunately deleted all tracked changes and cannot recover them. In this new version, I will ensure that all minimal changes are included as tracked changes.

186 - 189: the terms should be "fixed effect" and "random effect"

Changed as suggested.

191: Edit: linear

Changed as suggested.

194: why was replicate not included as a random effect here when it was above? Also, can you please clarify "interaction effects"? Which interactions did you include?

Changed as suggested. Explained above and in methodology. Hatchability models with random effect variable were poor fitted and validated. The interactions for hatchability were a four-way (season, blood source, cycle and replicate)

207-208: I'm not sure what you mean by "aimed to achieve"? Weren't you doing this after you conducted the experiments, so wouldn't this be determining the power of your model (post-hoc power analysis)? Also, I think you should provide the parameter estimates that were used (e.g. effect size - did you use the effect size you estimated across the 3 replicates?).

Changed as suggested.

214-215: this should be reworded to acknowledge that this is estimated for the given effect size; for example, something like "This sample size was sufficient to detect the observed effect with a statistical power of 0.8" or something along those lines (unless I am misunderstanding how you conducted this test).

Changed as suggested.

246. Abbreviate Culex

Changed as suggested.

253-255: This sentence isn't clear. What do you mean by mixed? Also, the season really seemed to mainly impact the fitness of mosquitoes fed on mouse blood and here the way it is phrased seems to indicate that season has an impact on the fitness of those fed with chicken blood.

Changed as suggested.

258-260: You stated your hypothesis as the relative fitness shifting between seasons, but this statement about the expectation is different from your hypothesis stated earlier. Please clarify.

You are right. Thank you for noting this. It was changed as suggested.

263-266: I also don't understand this sentence; what does the first half of the sentence have to do with the second?

Changed as suggested.

269-270: This doesn't align with your observation exactly; you say first AND second are generally most productive, but you observed a drop in the second. Please clarify this.

Changed as suggested.

280: I suggest removing "as same as other studies"; your caveats are distinct because your experimental design was unique

Changed as suggested.

287: you shouldn't be looking for a "desired" effect; I suggest removing this word

Changed as suggested.

288: It wasn't really a priori though, since you conducted it after your first replicate (unless you didn't use the results from the first replicate you reported in the original drafts?)

It was a posteriori. Changed as suggested.

290: Why is 290 written here?

It was a mistype. Deleted as suggested.

291-298: The meaning of this section of your paragraph is not clear.

Improve as suggested.

304-313: This list of 3 explanations are directed at different underlying questions. Explanations 1 and 2 are alternative explanations for why host switching occurs if not due to differences in fitness. This isn't really an explanation of your results so much as alternative explanations for a previously reported phenomenon. And the third is an explanation for why you may not have observed the expected effect. I suggest restructuring this to include the fact that Argentinian quinqs may not host switch as part of your previous list of caveats. Then you can include your two alternative explanations for host switching as a possible future direction (although I would say that it is really just one explanation because "vector biology" is too broad of a statement to be testable). Also, you haven't discussed possible explanations for your actual result, which showed that mosquito fitness decreased when feeding on mouse blood in autumn conditions and in the second gonotrophic, while those that fed on chicken did not experience these changes. Why might that be?

The discussion was restructured to include all these suggested changes. Additionally, it was also discussed some possible explanations of our results.

315-317: This statement is vague without a direct explanation of how this will provide insight. I suggest removing or providing an explanation of how this provides insight to transmission and forecasting.

Changed as suggested.

319-320: According to eLife policy, all data should be publicly available. From guidelines: "Media Policy FAQs Data Availability Purpose and General Principles To maintain high standards of research reproducibility, and to promote the reuse of new findings, eLife requires all data associated with an article to be made freely and widely available. These must be in the most useful formats and according to the relevant reporting standards, unless there are compelling legal or ethical reasons to restrict access. The provision of data should comply with FAIR principles (Findable, Accessible, Interoperable, Reusable). Specifically, authors must make all original data used to support the claims of the paper, or that is required to reproduce them, available in the manuscript text, tables, figures or supplementary materials, or at a trusted digital repository (the latter is recommended). This must include all variables, treatment conditions, and observations described in the manuscript. The authors must also provide a full account of the materials and procedures used to collect, pre-process, clean, generate and analyze the data that would enable it to be independently reproduced by other researchers."- so you need to make your data available online; I also understand the last sentence to indicate that code should be made available.

Data and code will be publicly available.

Table 1: it is notable that in replicate 2, the autumn:mouse:gonotrophic cycle II fecundity and fertility are actually higher than in the summer, which is the opposite of reps 1 and 3 and the overall effect you reported from the model. This might be worth mentioning in the discussion.

Mentioned in the discussion as suggested.

Tables 1 and 2: shouldn't this just be 8 treatments? You included replicate as a random effect, so it isn't really a separate set of treatments.

This table reflects the output of the whole experiment, that is why it is present the 24 expetiments.

Figure 3: Can you please clarify if this is showing raw data?

Changed as suggested.

Note: grammatical copy editing would be beneficial throughout

Grammar was improved as suggested.